# Do Countries with Similar Levels of Corruption Compete to Attract Foreign Investment? Evidence Using World Panel Data

**Luisa Alamá-Sabater** [1], **Teresa Fernández-Núñez** [2], **Miguel Ángel Márquez** [2] **and Javier Salinas-Jimenez** [3,*]

1   Department of Economics and Local Development Institute (IIDL), Universitat Jaume I, Campus del Riu Sec, 12071 Castelló de la Plana, Spain; alama@uji.es
2   Department of Economics, Faculty of Economics and Business Administration, University of Extremadura, 06071 Badajoz, Spain; teresafn@unex.es (T.F.-N.); mmarquez@unex.es (M.Á.M.)
3   Departamento de Economía y Hacienda Pública, Universidad Autónoma de Madrid, 28049 Madrid, Spain
*   Correspondence: javier.salinasj@uam.es

**Abstract:** This paper examines whether foreign direct investment in one country helps to increase foreign investment in other countries with a similar degree of corruption. Our estimates are based on an unbalanced annual panel of 164 countries over the 2005–2015 period. Using spatial econometric techniques, our main findings reveal that foreign investment in one recipient country is complementary to that in countries with similar levels of corruption. Furthermore, our results point to the existence of different circuits of foreign direct capital among countries that are determined by corruption similarity. These results suggest important policy implications for countries aiming to attract foreign investment.

**Keywords:** foreign direct investment; sustainable development; corruption similarity; spatial econometrics

## 1. Introduction

In the last two decades, there has been a reestablished interest in examining the driving forces of foreign direct investments (FDI). This renewed interest goes with recognition of the key role of FDI to enhance sustainable development, which is the main focus of the United Nation's 2030 Agenda. In 2015, the United Nations (UN) General Assembly set 17 Sustainable Development Goals (SDGs) designed to promote economic, environmental, and social sustainable development at the national and international level. The contribution of FDI to the achievement of SDGs has been largely acknowledged through its impact on international trade, economic growth, poverty reduction, or environmental sustainability [1–3]. The role of governance in this context becomes then twofold. On the one hand, the links between governance and sustainability have been largely analyzed, showing that good governance in terms of the rule of law, bureaucratic quality, and control of corruption show positive and significant effects on sustainable development [4,5]. On the other hand, it is well established that good governance plays a significant role on economic development and FDI flows [6], which are seen as the principal means to finance sustainable development [7]. Governments can support investments, but the involvement of the private sector becomes essential and, in the context of globalization, the role multinational enterprises (MNE) with activities abroad via FDI is emphasized for achieving sustainable development [8]. Policies both at home and in the host country should hence be designed to enhance FDI, so the need to better understand the determinants of FDI flows and the circuits of foreign capital among countries in order to promote a better integration of local and foreign business leading to a more sustainable development.

Traditionally, the determinants of FDI flows in specific locations have been explained by economic factors. However, the new extensions of the international trade theory, the so-called institutional approach, and the development of spatial econometric tools, have recently led to consider additional determinants for FDI [9–11]. Looking at these new developments of the literature, several empirical studies highlight the role of spatial interdependence in FDI decisions across host countries [12–14]. These studies examine whether FDI flows in a given country are complementary or substitute to those in geographically near countries. In this regard, it is worth noting that no conclusive outcomes have been reached [11,15].

Furthermore, there is a broad literature that focuses on the influence of corruption on FDI. One strand of this research has emphasized the effects of corruption in the host country on FDI flows [16–19], with the majority of empirical studies finding a negative impact of the recipient country's corruption on FDI. A second strand of this literature has focused on the join effects of home and host country corruption (i.e., corruption distance) on bilateral FDI flows. Whereas a few authors have showed that the distance in corruption levels between the origin and destination countries has a negative impact on the attraction of FDI [20–23], others, reached opposite results [24]. Recently, new researches highlight the importance of home country corruption in FDI flows [25–28], suggesting that the direction of the institutional distance matters [29].

This paper expands prior FDI works on both spatial interdependences and corruption distance impact on FDI. The contribution of this study to the existing literature comes from the determination of patterns of substitution or complementary in FDI across countries sharing similar degrees of corruption. In particular, this paper aims to provide answers to the following questions: Does foreign investment in one country help to raise foreign investment in other countries showing similar levels of corruption? Does proximity in corruption levels promote circuits of investment flows among countries? In so doing, it appears to be FDI dependence across host countries derived from their similarity in corruption levels. To our best knowledge, previous empirical studies have failed to take into account these dependences. Our main hypothesis is that investments into a given country depend on investments received by other countries with similar levels of corruption, once other characteristics are controlled for.

More specifically, the present study makes several contributions. First, a channel for FDI interdependence related to the externalities derived from the corruption similarity among host countries is disclosed. Empirical testing is performed to show not only the existence of competition or complementarity among different locations sharing similar corruption levels, but also to determine whether investors use corruption similarity to establish circuits of FDI among groups of countries. To this end, a spatial lag model (SAM) framework is considered, applied to a spatially unbalanced annual panel with 164 countries for the period from 2005 to 2015. Second, relative to previous studies, a wider sample of countries is used to improve the accuracy of the estimates. Although in the FDI determinants literature, it is usual to select a sample of countries that are somewhat homogeneous [14], some authors point out that this sampling may distort the cumulative knowledge in this field [11,30].

This paper is organized as follows. Section 2 describes the theoretical background and the economic intuition of the study. Section 3 introduces the data and provides an exploratory analysis. Section 4 presents the econometric specification of the model used to show empirical evidence of the proposed hypothesis. In Section 5, the empirical analysis is described, highlighting some econometric issues and discussing the empirical results. Section 6 concludes.

## 2. Theoretical Background

Corruption is a complex phenomenon that can influence foreign investment decisions in different ways [28]. The main strand of the literature on corruption and FDI points to corruption as a 'grabbing hand' that negatively affects FDI inflows by conditioning FDI location decisions, creating an unfavorable environment for multinational enterprises because of the additional costs and the uncertainty and risks of operating in a shady business climate [18,20,31]. Moreover, corruption may show a negative impact on key determinants of FDI, such as economic growth, productivity, and the quality of

infrastructures [16]. However, an alternative view sees corruption as a 'helping hand' that could 'grease' the economy in the context of inefficient economic systems and weak regulatory frameworks [32]. In this case, corruption could serve to speed money and attract FDI, expediting decisions and avoiding government requirements, hence acting as a mechanism for deregulation and increasing the efficiency of multinational firms [6,17,19]. The empirical literature provides mixed evidence about the effects of host country corruption on FDI attractiveness, although a negative impact usually prevails over the positive one.

FDI flows are also affected by the join effects of the levels of corruption in the home and host countries. Multinational enterprises develop knowledge, skills, and use business practices based on their domestic experience and their home corruption environment [33]. These skills may represent a competitive advantage and may be used efficiently to invest in recipient countries with similar corrupt environments [28,34,35]. In this sense, some authors have shown that differences in corruption levels between the origin and destination countries have a negative effect on bilateral FDI [21–23,28,36]. The rationale for the adverse impact on FDI bases on the hurdles that distance could generate with regard to the transfer of knowledge [37], as well as the capability and extra-cost to understand the values and rules of the foreign market [22,38,39]. However, contrary to expected conjectures, a few authors, such Bellos and Subasat [24] and Hausman and Fernández-Arias [40], conclude that less corrupt countries tend to invest more in higher corrupt economies. In order to resolve these ambiguous findings, recent studies highlight the need to consider the asymmetric effects of distance on foreign investment locations [29,41], with some authors arguing that the effects of corruption in the host country on FDI depend on the level of corruption in the home country [25–28].

Bearing all these considerations in mind and deepening the approach based on measuring institutional distance, this paper focuses on the effects of corruption on FDI in different host countries sharing similar corruption environments. Foreign investors can have different perceptions of their investments risks in host countries and also different tolerance towards corruption [27]. These differences may come from their own expertise reached in doing business in corrupt environments—at home or abroad. In this sense, following the organizational learning theory, this experiential learning might serve foreign investors as a guide for their behavior. They could perceive that host countries with similar levels of corruption have a similar administrative or institutional culture, similar levels of governance development, the same way of doing business, and/or similar indirect costs associated with uncertainty. In addition, corruption similarity could provide close conditions to facilitate the transference of knowledge and skills and offer similar opportunities to connect with local business. Consequently, multinationals could take advantage of their abilities to adapt their investments in these nearby countries. Therefore, the decision to invest in a foreign destination may be influenced by what is happening in other countries with similar degrees in corruption. In other words, it could be assumed the presence of cross-country correlation of FDI flows across recipient countries nearby in levels of corruption. Following Alamá-Sabater et al. [42], one could think about these cross-country correlations as interdependences, so corruption similarity could be considered as a new channel for interdependence of FDI flows.

Moreover, the potential presence of spatial dependence in the FDI location across host countries has been mainly related to the geographical proximity between economies. Since the beginning of this century, the new extensions of the theory of trade and multinational firms, such as the theory of export platform FDI [43,44] and the theory of complex vertical FDI [45,46], along with the development of spatial econometric tools in FDI studies have led to consider the role of the geographical component on the determinants of FDI location decisions. Accordingly, several empirical studies have incorporated spatial dependence as an important FDI driver [12,13,46–49], taking account of two forms of spatial interdependences: Surrounding market potential (related to the size of markets close to the host country and measured by the weighted GDP of neighboring markets) and FDI spillovers (referred to as the capital inflows to neighboring countries and measured by the weighted FDI in neighboring countries). FDI flows in a given country could then depend not only on its domestic conditions, but also on those

of its neighboring countries, including their FDI inflows [12,13]. In econometric terms, geographic proximity has been modeled as a source of cross-country dependence (geographic proximity has been delimited by different tools like, among other, contiguity measures, some physical distant function, a choice of k-nearest neighbors, or by some combination of these options). The empirical evidence has yielded mixed results about the significance of the spatial connectivity between countries and about the nature of this spatial interdependence, with results being very sensitive to the selected sample of countries.

In recent years, the use of weight matrices based solely on spatial location has been widely criticized [50]. Thus, nowadays, alternative forms of proximities beyond just geographic space, based on sociocultural or institutional factors, are being adopted [51–54]. Nevertheless, they are focused on the role played in economic growth instead of in the attractiveness of foreign investments. Empirical studies centered on non-geographic distance as channel of spatial dependence in the context of FDI location are very scarce [42,55,56]

To the best of our knowledge, none of the previous researches take into account a dimension of neighborhood based on the concept of corruption similarity. This paper is then a first attempt to reveal another additional dimension of FDI interdependence (other than geographical proximity) in a multi-country framework, exploring whether corruption similarity might be a driver of the spatial interdependence of FDI flows among host countries.

## 3. Variables and Data Description

The empirical analysis bases on an unbalanced panel of inward FDI data in 164 host countries for the period 2005–2015. Data used in this paper require some explanation. First, in line with other studies [13,35,57,58], flow data are preferred to stocks to analyze the FDI entry decision, as it shows how MNE investments in a given location change from year to year. With stock data, changes in the entry or in the number of firms might not be clear and appreciable when they take place for a huge accumulated base value [11,58]. Second, for the period under analysis, 80 observations with negative values for FDI inflows were found in the database (representing 4% of the sample). In order to avoid missing values when taking logs, some authors [13] transform negative values into a very small number, almost zero. Other papers consider the missing values to be equal to zero [46]. In the present paper, the negative values were removed, the model being hence explained only by positive FDI inflows. This decision is reinforced by the facts that these values are associated with exceptional situations and that the focus of the study is on decisions on investments' location, not on disinvestment. Third, this *modus operandi* requires the use of an adequate strategy, so in order to maintain information data for relevant countries showing negative values in some years (for example, China or Japan), an unbalanced spatial panel data approach was adopted.

The dependent variable in our study is the real FDI inflows (FDI). Data on nominal FDI flows in US dollars come from UNCTAD FDI database and were deflated by the GDP deflator (2010 = 100) provided by the World Bank's World Development Indicators (WDI).

With regards to the independent variables, our main variable of interest is the level of corruption in the recipient countries. To analyze corruption, researchers usually use survey data as a basis, which captures the perceived existence of corruption. One of the most accepted indices in the empirical literature is the Corruption Perception Index published by Transparency International every year since 1995. This index measures the perception of experts and business executives about corruption in a country's public sector and is the one used in this study (denoted as CI). It should be noted that this index range from 0 (high corruption) to 100 (low corruption), so in fact it could be seen as a variable of 'control of corruption'. Although there is the alternative to transform this index, it is usual in the empirical literature on corruption to use this index without any transformation. This last option is followed in this paper to facilitate comparisons with other works, but bearing in mind that a positive coefficient would indicate that a lower (higher) level of corruption will enhance (worsen) FDI flows. Moreover, following the conventional literature on the determinants of FDI, a set of control variables

reflecting host country characteristics is also considered. The choice of these independent variables is also conditioned by data availability. Related to economic and institutional factors, GDP *per capita* (GDPpc) is considered as a measure of potential consumption. Population (POP) is used to measure the market size. Trade openness (OPEN) is seen as a key driver that influences FDI decisions; it is calculated as the share of total exports and imports in GDP. The quality of human capital (HK) is proxied by skilled workforce and is calculated as gross secondary school enrolment in the recipient country. Inflation rate (INFL) is used to control the overall economic stability in the recipient country and is obtained from the annual growth rate in the GDP implicit deflator. The percentage of urban population (URBPOP) is taken as a proxy for urbanization, considering that a high degree of urbanization tends to foster FDI inflow since it usually goes with a better infrastructure quality and a higher concentration of consumers. Finally, economic freedom (EF) in the recipient country is measured by the economic freedom index, which was gathered from the Heritage Foundation. Table 1 presents the complete list of variables, a summary of definitions, their data sources, and the expected impact of the independent variables on inward FDI.

**Table 1.** Description of variables and sources.

| Variable | Description | Source | Expected Sign |
|---|---|---|---|
| Ln *FDI* | Logarithm of total FDI inflows in a host country in constant 2010 US dollars (deflated by GDP deflator) | UNCTAD WDI | |
| Ln *GDPpc* | Logarithm of Gross Domestic Product in constant 2010 US dollars divided by population | | (+) |
| Ln *POP* | Logarithm of total population | WDI | (+) |
| Ln *OPEN* | Logarithm of trade openness (exports plus imports) as share of GDP | WDI | (+) |
| Ln *HK* | Logarithm of Secondary School Enrollment (gross %) | WDI UNESCO | (+) |
| *INFL* | Annual growth rate of the GDP implicit deflator | WDI | (-) |
| Ln URBPOP | Logarithm of urban population (%) | WDI | (+) |
| Ln *EF* | Logarithm of Economic Freedom Index. This index varies between 0 (non-existent) and 100 (excellent) | Heritage Foundation | (+) |
| Ln CI | Logarithm of Corruption Perception Index. This index discloses the degree to which corruption is perceived to be among public officials. This index ranges between 0 (high level of corruption) and 100 (low level) | Transparency International | (+) |

Table 2 offers the descriptive statistics of these variables, showing that countries considered in the analysis display a broad variation in the sample period with regard to the macroeconomic and institutional variables. This fact in turn is likely to condition their expected performance in terms of FDI attractiveness.

**Table 2.** Summary statistics.

| | Mean | Std. dev. | Min. | Max. | Median |
|---|---|---|---|---|---|
| *FDI* | 9389.01 | 25,019.84 | 0.04 | 349,581.00 | 1288.13 |
| *GDPpc* | 1,0557.04 | 15,520.25 | 140.76 | 86,117.99 | 3428.17 |
| *POP* | 4,2565.26 | 148,608.50 | 70.54 | 1,376,049.00 | 9556.50 |
| *OPEN* | 92.69 | 57.27 | 0.17 | 455.42 | 81.57 |
| *HK* | 76.78 | 28.82 | 9.91 | 164.8 | 85.11 |
| *INFL* | 6.39 | 9.38 | −29.69 | 121.74 | 4.40 |
| *URBPOP* | 56.78 | 23.11 | 8.45 | 100.00 | 57.48 |
| *EF* | 59.81 | 11.45 | 1.00 | 90.10 | 59.6 |
| *CI* | 41.46 | 20.47 | 8.00 | 97.00 | 34.00 |

## 4. Econometric Specification

To study the main determinants of inward FDI at the world level, the first step stands on a conventional econometric specification that takes into consideration the classical FDI determinants used in the literature. The baseline model for a panel of $N$ observations over $T$ periods of time can be written as follow:

$$lnFDI_{it} = \beta X_{it} + \epsilon_{it} \tag{1}$$

where the dependent variable $lnFDI_{it}$ is a column vector of size *N x 1* that shows the logarithm of the flow of FDI that every country $i$ receives from the rest of the world in year $t$. $X_{it}$ denotes an *n x k* matrix of independent variables (see the definition of these $k$ variables in Table 1) while $\beta$ is a vector of unknown parameters. $\epsilon$ is an error term that is independently normally distributed.

To account for the existence of spatial interdependence of FDI among our representative sample of countries in the world, a spatial lag model (SLM) was specified [12–14,59]. The panel model in Equation (1) is extended by introducing a spatially lagged dependent variable and a spatially lagged GDP (surrounding market potential). This specification can be expressed as follows:

$$lnFDI_{it} = \lambda W_g lnFDI_{it} + \sigma W_g lnGDP_{it} + \beta X_{it} + \epsilon_{it} \tag{2}$$

where $\lambda$ is the parameter to be estimated and $W_g$ $ln$ $FDI$ denotes the spatial lag of the dependent variable in the model of FDI. $W_g$ represents the geographical weighting matrix for the whole panel, and it is block-diagonal with typical block $W_{Nt}$:

$$W_g = \begin{pmatrix} W_{Nt} & 0 & 0 \\ 0 & W_{Nt} & 0 \\ 0 & 0 & W_{Nt} \end{pmatrix} \tag{3}$$

$W_g$ is made up of T×T matrices, $T$ being the number of years in the sample. In our analysis, as the panel database is unbalanced [17], $W_{Nt}$ has different dimensions for each year (specifically ($N_t \times N_t$)).

The $W_{Nt}$ matrices were constructed following a geographical criterion based on the common border between countries. Note that all countries have at least one neighbor. In order to avoid the case of isolated countries, that is, those without any common physical border (for instance, Japan or Australia or bordering countries of those removed from the sample), the nearest country has been considered as the neighboring country. The typical entry in row $i$ and column $j$, $w_{ij}$, is defined as follows:

$$w_{ij,t} \begin{cases} 1 \; if \; country \; i \; and \; country \; j \; share \; a \; common \; border \\ 0 \; if \; i \; and \; j \; do \; not \; share \; a \; common \; border \; or \; i = j \end{cases}$$

The different block-diagonal matrices were row-standardized by dividing each row of each matrix by the respective row sum.

$W_g lnGDP$ is a proxy variable for market potential, and σ is the market potential coefficient. This variable is calculated as the distance weighted average of GDPin constant 2010 US dollars of all recipient countries where $j \neq i$. Following [12], the combination of expected signs of both spatial lag ($\lambda$) and market potential coefficients (σ), could reveal various FDI strategies linked to specific theories (e.g., pure horizontal, export platform, pure vertical, and complex vertical specialization of agglomeration effects).

To further explore whether FDI in a given host country depends on the FDI flowing in other countries sharing a similar level of corruption, a new dimension of neighborhood based on the concept of corruption similarity is considered. To capture this potential channel of FDI interdependence, according

to Alamá-Sabater et al. [42], the following fuzzy metric (*Corruption Similarity Index* $- SCI_{ij}-$) related to the level of corruption between peer countries was constructed:

$$SCI_{ij} = \frac{min\{corruption\ index\ (i),\ corruption\ index\ (j)\} + 1}{max\{corruption\ index\ (i),\ corruption\ index\ (j)\} + 1} \tag{4}$$

where $SCI_{ij}$ represents a continuous measure of similarity in corruption levels between FDI host country $i$ and $j$. $SCI_{ij}$ is defined between (0,1), decreasing with a larger difference between the corruption indices of host countries and increasing when two countries have similar levels of corruption; the interdependence will be higher when the level of corruption of the FDI hosts countries is more similar. This indicator allows us to extend Equation (2) to incorporate a new dimension of neighborhood based on the corruption similarity among countries (the consideration of different weight structures to model dependence among spatial units is not a new approach [60]:

$$lnFDI_{it} = \lambda W_g ln\ FDI_{it} + \sigma W_g lnGDP_{it} + W_c\ ln\ FDI_{it} + \beta X_{it} + \epsilon_{it} \tag{5}$$

where $\gamma$ is the parameter to be estimated to account for the effect of the similarity in FDI among countries, and $W_c\ ln\ FDI$ is the similarity lag of the dependent variable (FDI) (Socioeconomic distance matrices have been used in the literature to capture interdependences among spatial units [56,61,62]. $W_c$ represents the similarity weighting matrix for the whole panel, its block-diagonal being built by the matrices $W_{c,Nt}$:

$$W_c = \begin{pmatrix} W_{c,Nt} & 0 & 0 \\ 0 & W_{c,Nt} & 0 \\ 0 & 0 & W_{c,Nt} \end{pmatrix} \tag{6}$$

As noted above, since the panel database is unbalanced, $W_{c,Nt}$ is made up of T × T matrices, T being the number of years in the sample. $W_{c,Nt}$ has different dimensions for each year ($N_t \times N_t$).

Taking into account the corruption similarity index in Expression (4), different $W_{c,Nt}$ matrices are constructed (according to the corruption indices of each year). Each element of $W_{c,Nt}$ in row $i$ and column $j$, $w_{ij}$ was defined as the value of the SCI (Expression 4) for countries $i$ and $j$.

In order to illustrate the use of the similarity lag proposed in Equation (5), a simple numerical example is presented, offering basic details about how to get this information. Let us assume that four countries exist: A, B, C, and D, with A and B having a similar low level of corruption, while C and D show a similar high level of corruption: A = 90, B = 85, C = 15, D = 10. By applying Expression (4), the similarity matrix (equivalent to $W_{c,Nt}$) can be obtained:

SCI (i,j) values from Table 3 represent the weights that will be used to calculate the similarity lag of FDI flows in each country. For example, the FDI inflow of country A (FDI$_A$) is related to the FDI flows in each of the other countries (FDI$_B$, FDI$_C$, FDI$_D$), weighted according to the peers' SCI. In this way, the channel of dependence on FDI inflows shows that investors consider the signal for investing in countries A, B, C, and D through the similarity lag (the equivalent elements of $W_c\ ln\ FDI_{it}$ from Equation (4)) as follows:

$$w_{c,A}\ ln\ FDI_A = 0.945 ln\ FDI_B + 0.176\ ln\ FDI_C + 0.121\ ln\ FDI_D$$

$$w_{c,B}\ ln\ FDI_B = 0.945 ln\ FDI_A + 0.186\ ln\ FDI_C + 0.128\ ln\ FDI_D$$

$$w_{c,C}\ ln\ FDI_C = 0.176\ ln\ FDI_A + 0.186\ ln\ FDI_B + 0.687\ ln\ FDI_D$$

$$w_{c,D}\ ln\ FDI_D = 0.121\ ln\ FDI_A + 0.128\ ln\ FDI_B + 0.687\ ln\ FDI_C$$

**Table 3.** Similarity Corruption Index (SCI) example.

| | SCI($i,j$) | | | |
|---|---|---|---|---|
| | **A** | **B** | **C** | **D** |
| **A** | 0 | 0.945 | 0.176 | 0.121 |
| **B** | 0.945 | 0 | 0.186 | 0.128 |
| **C** | 0.176 | 0.186 | 0 | 0.687 |
| **D** | 0.121 | 0.128 | 0.687 | 0 |

In this example, if the coefficient estimated for the parameter $\gamma$ in Equation (5) was significant and positive, a country's inward FDI would be influenced by FDI received by other countries showing similar levels of corruption (there would be evidence of a contagion effect). Conversely, if the estimation of $\gamma$ was significant and negative, there would be competition for FDI inflows between countries with similar levels of corruption. Figure 1 shows the similarity corruption index for the least and most corrupt countries in the world in 2015; that is, Denmark and Afghanistan, respectively.

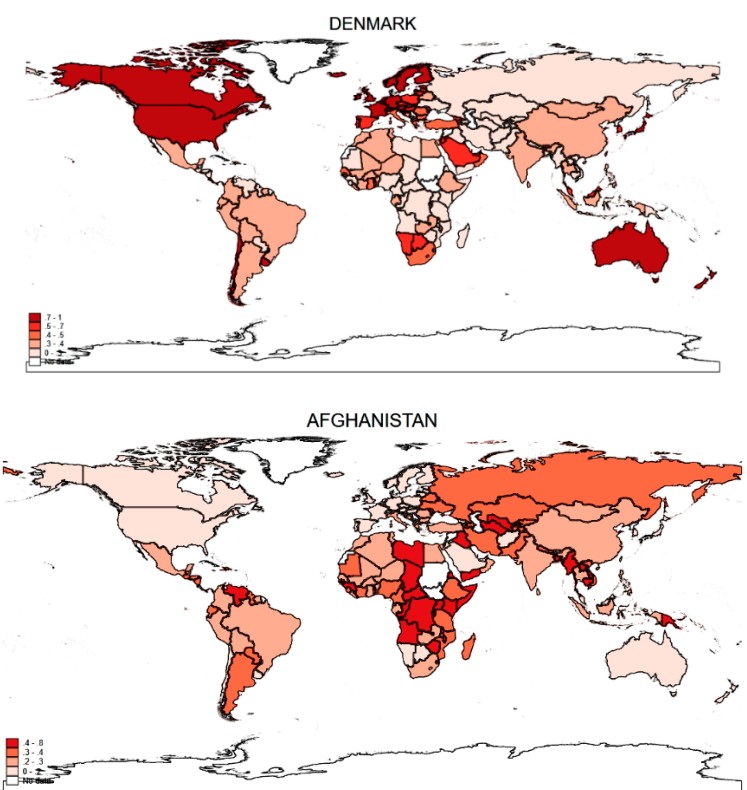

**Figure 1.** Similarity corruption index in Denmark and Afghanistan in 2015. Source: Authors' own contribution with data from Transparency International.

The channel of interaction proposed in this paper comes from the idea that changes in inward FDI to a country $i$ will influence the inward FDI to other countries $j$ ($j \neq i$) showing a similar level of corruption to that of country $i$. Only inward FDI to countries with similar levels of corruption will capture non-negligible effects of inward FDI explained by this concept of neighborhood (corruption similarity). As a matter of example, it seems more likely that FDI interdependences appear between Denmark and USA or Canada, that share similar levels of corruption (as shown in Figure 1) than between Denmark and countries with low values of peers' SCI, for example Afghanistan.

## 5. Econometric Issues and Empirical Results

The empirical analysis used to determine the significance of the two FDI interdependence channels (i.e., geography and corruption similarity) begins with an estimation of Equation (1) as a benchmark. The estimation of Equation (1) for our unbalanced panel of 164 countries over the period 2005–2015 will assess the influence of the traditional determinants of FDI. The next step will be to compare these results with our proposed extended specification in Equation (5).

The model selection strategy started with the ordinary least square (OLS) estimation of Equation (1) and the alternative of a panel fixed effects model. After the rejection of the OLS estimator, a test was run for the null hypothesis of a fixed effects model and the alternative of a two-way fixed effects model (fixed effects for time periods were added to the set of explanatory variables in the two-way model and, with respect to the dummy variable trap for $\eta_t$ (t = 2005, ... , 2015), the dummy relative to year 2005 was dropped). The test implied the rejection of the null hypothesis, leading to the conclusion that the data contain significant country effects as well as significant time effects. In a next step, fixed or random country effects were allowed, with the Hausman test showing that country-specific effects should be modeled as random [63]. In consequence, Column (1) of Table 4 presents the baseline results of Equation (1) coming out of the estimation of a random effects model with time effects, where robust standard errors were used. The signs of the estimated coefficients for most of the traditional control variables are positive and significant. The estimated coefficient for corruption shows a positive and significant effect (at the 10% level), a result that suggests that lower levels of corruption imply a weak advantage for FDI location decisions, supporting the theoretical hypothesis of corruption acting as a "grabbing hand" for international investors. Inflation also has an unexpected positive sign on FDI, but this effect is not significant.

**Table 4.** Determinants of foreign direct investments (FDI) flows (2005–2015).

| Variables | Baseline Model (Random Effects) | Two Channels (Random Effects) | Corruption Similarity Channel (Random Effects) | Geographic Channel (Random Effects) |
|---|---|---|---|---|
| $W_g Ln$ FDI | | 0.102 *** | | 0.091 *** |
| | | [0.006] | | [0.009] |
| $W_g LnGDP$ | | −0.175 ** | | −0.158 ** |
| | | [0.030] | | [0.043] |
| $WcLnFDI$ | | 0.504 *** | 0.586 *** | |
| | | [0.000] | [0.000] | |
| *Ln GDPpc* | 0.647 *** | 0.695 *** | 0.629 *** | 0.709 *** |
| | [0.000] | [0.000] | [0.000] | [0.000] |
| *Ln POP* | 0.790 *** | 0.771 *** | 0.790 *** | 0.774 *** |
| | [0.000] | [0.000] | [0.000] | [0.000] |
| *Ln OPEN* | 0.399 ** | 0.384 ** | 0.393 ** | 0.386 ** |
| | [0.015] | [0.012] | [0.012] | [0.014] |
| *Ln HK* | 0.610 ** | 0.529 * | 0.556 ** | 0.599 ** |
| | [0.026] | [0.056] | [0.042] | [0.036] |
| *INFL* | 0.058 | −0.048 | 0.011 | −0.003 |
| | [0.842] | [0.870] | [0.969] | [0.991] |
| *Ln URBPOP* | 0.482 * | 0.521 * | 0.511 * | 0.483 * |
| | [0.066] | [0.076] | [0.057] | [0.092] |
| *Ln EF* | 0.059 | 0.038 | 0.056 | 0.040 |
| | [0.694] | [0.804] | [0.705] | [0.801] |
| *Ln CI* | 0.400 * | 0.126 | 0.108 | 0.384 |
| | [0.096] | [0.656] | [0.710] | [0.108] |
| *Constant* | −8.690 *** | −11.384 *** | −11.556 *** | −8.885 *** |
| | [0.000] | [0.000] | [0.000] | [0.000] |

<div align="center">**Table 4.** *Cont.*</div>

| Variables | Baseline Model (Random Effects) | Two Channels (Random Effects) | Corruption Similarity Channel (Random Effects) | Geographic Channel (Random Effects) |
|---|---|---|---|---|
| Fixed time effects | Yes | Yes | Yes | Yes |
| Observations | 1724 | 1724 | 1724 | 1724 |
| Number of Countries | 164 | 164 | 164 | 164 |
| R-squared-overall | 0.730 | 0.721 | 0.730 | 0.722 |
| R-squared-between | 0.805 | 0.793 | 0.804 | 0.796 |
| R-squared-within | 0.0970 | 0.112 | 0.108 | 0.105 |
| Hausman test (Prob > chi2) | | 0.5356 | 0.1110 | 0.0159 |
| Wald test (SLM vs. OLS)(prob > chi2) | | 0.0000 | 0.000 | 0.0093 |
| Sargan-Hansen statistics *p*-value | | 0.3081 | 0.4966 | 0.1007 |
| First-stage F-statistics: WglnFDIit | | 18.70 | | 25.42 |
| First-stage F-statistics: WclnFDIit | | 1119.46 | 2221.31 | |

<div align="center">*** $p < 0.01$, ** $p < 0.05$, * $p < 0.1$ (*p*-value in brackets).</div>

The next step is to test the validity of the two channels of FDI interdependencies proposed in Equation (5), that is, among FDI host countries that share a common border (geographical proximity) and among FDI host countries that are similar in terms of corruption (corruption similarity). This model was first estimated via a two-stage least square estimator (2TSLS) with endogenous regressors [64,65], based on feasible instruments like H = ($X$, $WX$, $W^2X$) [66], where X represents the explanatory variables and W is the spatial weights matrix [67]. Spatial lags of the explanatory variables were used as proper instruments for $W_g$ *ln FDI* and $W_c$ *ln FDI* [61,66,68]. Following these authors, one strategy to deal with the endogeneity problem is to catch some source of variation correlated with related countries' investment but uncorrelated with the error term. As a result, weighted averages of the explanatory variables were used as instruments of the spatial lag variables ($W_gX$'s were used as instruments for $W_g$*ln FDI*, while the instruments for $W_c$ *ln FDI* were $W_cX$'s).

Column (2) of Table 4 presents the second-stage results of the instrumental variable estimation for specification (5). In accordance with the Hausman test, the model was estimated using a random effects specification (the random effects (RE) model was chosen based on the Breusch and Pagan Lagrange multiplier (LM) test). It should also be noted that, as in the model in Column (1), fixed effects for time periods were added to the set of explanatory variables and the model was estimated with robust standard errors. The overall validity of the instruments used in the regression is evaluated by the Sargan-Hansen test (or over-identifying restrictions test) as well as by the first-stage F-statistics to check the relevance of the instruments. Moreover, the variance inflation factors (VIFs) do not show any evidence of multicollinearity. The set of tests provided at the bottom of Table 4 reflects that our estimation strategy is adequate and robust. Overall, the coefficients for the control variables validate the outcomes obtained using the baseline model (Column 1), which are consistent with the extant literature: Host country's market size, potential consumption, trade openness, infrastructure quality, and skilled workforce have a positive and statistically significant impact on FDI. However, the coefficient for the surrounding market potential is negative and statistically significant at the 5% level. As the economic activity for surrounding markets increases, FDI will decreases. This outcome is unexpected and inconsistent with the theory, although similar to that obtained by Blonigen et al. [12] and Nwaogu and Ryan [14], suggesting an apparent FDI motivation puzzle at the world level.

As regards corruption, it should be noted that, in line with Henisz [69], there is no evidence of a significant relationship between corruption and FDI inflows. However, a positive and statistically significant impact of both channels of interdependences for FDI (geographical and corruption similarity) is found. This result suggests that FDI inflows to a given country are correlated not only with FDI flows in geographically close countries, but also with FDI flows in countries with similar levels of corruption, with this last channel of FDI interdependence being even more relevant. In particular, an

unexpected change in FDI inflows to a country drives to a similar change in other host countries with similar levels of corruption. Thus, although the level of corruption in a given host country does not have a significant effect on attracting FDI, our results show that corruption similarity does provide a signal to foreign investors, lending support to a new channel of FDI interdependence: The higher the investments received by a particular country, the higher the FDI in other host countries with similar corruption levels. It seems that foreign investors perceive that they would need similar abilities to adapt their investments to countries with similar levels of corruption.

Finally, each channel of interdependence is separately considered to test the robustness of the results. Column (3) offers the estimates when only the channel of corruption similarity is included, with this variable being positive and statistically significant at the 1% level; the result of complementarity in FDI inflows to countries which are close in corruption levels remains hence unchanged. In Column (4) the variable of corruption similarity among countries is dropped, considering only spatial interdependences in FDI between host countries sharing a common border. The spatially lagged dependent variable is statistically significant at the 1% level, suggesting complementarity relationships between FDI inflows to a given country and those received by its neighboring countries. It should also be noted that, as it was the case in the joint specification shown in Column (2), the estimated coefficient for geographical interdependence is smaller than that for interdependence in FDI across host countries with similar levels of corruption. All in all, the estimates in Column (2) remain robust when the potential channels of FDI interaction among countries are estimated separately. Furthermore, the results are also robust to alternative forms of spatial weights matrix; specifically, another binary matrix (k-nearest neighbors spatial weight matrices for k = 4 and 5) and an inverse distance weighting matrix were also tested, obtaining similar results to those presented in Column (2) (these estimates have been omitted in the paper for the sake of brevity; full results are available from the authors upon request).

In sum, evidence is found for FDI interdependences among host countries due to both similarity in levels of corruption and geographical proximity, with corruption similarity showing a greater effect. Although the level of corruption does not have a significant impact on the attraction of FDI, corruption similarity provides a signal to foreign investors, so corruption similarity among countries provides a circuit where investors could potentially find analogous characteristics and *modus operandi* when doing business.

## 6. Conclusions and Final Remarks

Understanding the role played by corruption in fostering FDI inflows is a question of concern since it could have major implications for governments when trying to attract FDI. The objective of this study was to analyze how the attractiveness of a given country for foreign investors is affected by the investments received by other countries sharing similar levels of corruption, once other characteristics are controlled for. It should be noted, however, that some limitations to this study derive from the nature of the aggregated FDI data. Future studies should take into account whether the impact of corruption similarity as a channel for spatial dependence of FDI location decisions vary with the sectoral composition of FDI. Addressing this issue is an important and timely task, since FDI in different sectors might show very different connections to the rest of the economy.

Corruption is widely believed to negatively impact on foreign investments (Godinez & Liu, 2018), but our findings show that the effects of corruption on FDI attractiveness are not significant at the world level. Otherwise, our results show that corruption similarity provides a signal to foreign investors, lending support to a spatial interaction for FDI location. As a result, inward FDI flows in a country not only depends on its own location advantages (such as market size, trade openness, infrastructure quality, skilled workforce), but also on the flows received by geographically close countries and on the FDI inflows to countries with similar levels of corruption.

This paper contributes to previous literature by proposing and testing corruption similarity as a channel for spatial dependence of FDI location decisions. Our empirical results show that inward FDI to one country depends on FDI inflows to other countries with similar levels of corruption (*ceteris*

*paribus*). This reflects that foreign investment in a country is complementary to that in countries that are close in corruption levels, which implies that FDI flows in a group of countries sharing similar corruption backgrounds generate a contagion effect to other countries belonging to this group. These results point to the existence of different circuits of FDI at the world level, which are determined by the degree of corruption similarity among countries. This means that international investors in less corrupt countries could avoid foreign investments in countries with higher levels of corruption. In contrast, existent foreign investment in highly corrupt countries might facilitate FDI toward other countries showing underdeveloped institutions and governance, and consequently with a high level of corruption. The evidence of this channel would bring to light some restrictions in FDI flows: Not all countries might be able to attract foreign capital with the same accessibility from anywhere they want, since different levels of corruption deter inward FDI. An additional contribution of this paper refers to gaining new insights into the relevance of geography for capital flows by considering a large sample of countries with heterogeneous characteristics. This analysis shows that spatial dependences (neighboring effects) matter for FDI location decisions not only when a homogeneous area is considered, but also at the world level. FDI activities across adjacent host countries are complementary, as it happens with FDI flows across countries sharing similar institutions.

The policy implications of these findings arise when viewed in the context of the world competition to capture inward FDI. Actions targeted at attracting inward FDI could be more effective if they are implemented in coordination with other countries sharing a similar institutional background. Any action to improve the FDI attractiveness in a given country, such as reducing the level of corruption, could fail to generate additional FDI inflows if tackled in isolation. In this context, policy-makers should consider that healthy and stable economic conditions provide a competitive advantage for foreign investors, but also that fostering FDI inflows requires a common climate of doing business with geographically and institutionally nearby countries. Policy-makers should hence be concerned about implementing economic policies that promote trade and investment agreements among close countries since nationally tailored policies could fail to increase FDI inflows when FDI interdependences across countries are at play.

**Author Contributions:** Conceptualization, L.A.-S., T.F.-N., M.Á.M. and J.S.-J.; methodology, L.A.-S., T.F.-N., M.Á.M. and J.S.-J.; software, L.A.-S., T.F.-N., M.Á.M. and J.S.-J.; validation, L.A.-S., T.F.-N., M.Á.M. and J.S.-J.; formal analysis, L.A.-S., T.F.-N., M.Á.M. and J.S.-J.; investigation, L.A.-S., T.F.-N., M.Á.M. and J.S.-J.; resources, L.A.-S., T.F.-N., M.Á.M. and J.S.-J.; data curation, L.A.-S., T.F.-N., M.Á.M. and J.S.-J.; writing-original draft preparation, L.A.-S., T.F.-N., M.Á.M. and J.S.-J.; writing—review and editing, L.A.-S., T.F.-N., M.Á.M. and J.S.-J.; visualization, L.A.-S., T.F.-N., M.Á.M. and J.S.-J.; supervision, L.A.-S., T.F.-N., M.Á.M. and J.S.-J. All authors have read and agreed to the published version of the manuscript.

**Funding:** This research received no external funding.

**Acknowledgments:** Luisa Alamá-Sabater acknowledges the financial support of the Spanish Ministerio de Economía y Competitividad (ECO2017-85746-P). Miguel A. Marquez acknowledges the financial support of the Spanish Ministerio de Ciencia e Innovación (Ref. PID2019-109687GB-100).

**Conflicts of Interest:** The authors declare no conflict of interest.

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
