# Peer review of "Do Countries with Similar Levels of Corruption Compete to Attract Foreign Investment? Evidence Using World Panel Data"

_sustainability, doi:10.3390/su12156194_

Round 1

Reviewer 1 Report

From my point of view, this paper is well written and structured and addresses an interesting topic. Additionally, the methodology seems adequate to the projected aim. There are, however, a few points that should have to be taken into account. In particular:

1) To facilitate result's interpretation and make the reading more intuitive, I would rather transform the corruption indicator into the (100 - CI) form: To obtain 0 (low corruption) and 100 (high corruption). More intuitive to have a negative (expected) coefficient indicating that increased corruption worsens FDI flows.

2) Unnecessary repetition of equation (5) on page 7 (?).

3) Figure 1 : to facilitate reading, I would place the title on top of the figure. In addition, I would place the figure on a single page instead of splitting it in two.

4) Minor formatting checks (spacements, etc) would be welcome.

Summarizing, I think this to be a nice paper with potential improvement taking into account the above comments.

Author Response

Please find attached the revised version of our manuscript (“Do Countries with Similar Levels of Corruption Compete to Attract Foreign Investment? Evidence Using World Panel Data”) where we have taken into account the comments made during the referee process.

In particular, the following changes have been introduced:

1) It is usual in the empirical literature on corruption to use the CI index without any transformation, so we followed this option to facilitate comparisons with other works. However, we agree with the referee on his/her remark, so we have introduced some comments to clarify this point (see footnote 6).

2) The repetition of equation (5) has been solved.

3) Originally, the title of figure 1 was on the top. We think that this point refers to the formatting rules of the journal.

4) A careful spelling and editing check have been carried out throughout the paper.

We hope that these changes provide a satisfactory answer to the comments arisen in the referee process. We would also like to thank the reviewers for all comments and suggestions.

Reviewer 2 Report

Some corrections related to cited papers/authors:

  • row 96: Egger & al., 2005
  • page 9: in the footnote citacion, „Hausman, 1978” it's not found in References
  • row 156: correct name Blanc-Brude F
  • the paper no. 64(in References) is not cited in the text

The row 351 must be in the same page with Table 4.

Though the findings are clear and appropriate, I suggest, for the Conclusion part of the paper, a better exploitation of analyzes results and a short specification of the research limits.

Author Response

Please find attached the revised version of our manuscript (“Do Countries with Similar Levels of Corruption Compete to Attract Foreign Investment? Evidence Using World Panel Data”) where we have taken into account the comments made during the referee process.

In particular, the following changes have been introduced:

1) All the references have been checked and corrected.

2) Formatting has also been checked.

3) The conclusions have been revised to take account of the suggestions made by the reviewer as regards the exploitation of results and comments on the limits of the research.

We hope that these changes provide a satisfactory answer to the comments arisen in the referee process. We would also like to thank the reviewers for all comments and suggestions.

Reviewer 3 Report

Dear authors,

your research is very interesting. I have only a few suggestions:

  • try to explain the connection between corruption and FDI in the context of helping hand vs grabbing hand hypothesis 
  • sustainable development has four dimensions and good governance (where we can put the control of corruption) is surely very important. Try to explain this fact in your introduction.
  • you should write in the third person

All the best with your paper.

Best regards

Author Response

Please find attached the revised version of our manuscript (“Do Countries with Similar Levels of Corruption Compete to Attract Foreign Investment? Evidence Using World Panel Data”) where we have taken into account the comments made during the referee process.

In particular, the following changes have been introduced:
1) The connection between corruption and FDI has been emphasized in the context of the ‘helping hand’ vs the ‘grabbing hand’ hypotheses, as suggested by the reviewer.

2) The introduction has been revised in line with the suggestion of the reviewer to better explain the role of good governance (and the control of corruption) in the context of sustainable development.

3) A careful spelling and editing check have been carried out throughout the paper and it has been completely rewritten in third person.

We hope that these changes provide a satisfactory answer to the comments arisen in the referee process. We would also like to thank the reviewers for all comments and suggestions.